# Retrieval as Reasoning: Learning to Select and Generate with LLMs

## Abstract

Retrieval-Augmented Generation (RAG) (Lewis et al., 2020) has become a practical solution for addressing hallucination in large language models (LLMs) by conditioning responses on retrieved documents. However, existing RAG systems face two major limitations: (1) retrieval objectives are often misaligned with the downstream generation task, leading to irrelevant documents harmful to the generation; (2) concatenating many retrieved documents into long prompts strains model capacity and introduces positional biases that degrade performance. To overcome these issues, we propose a unified framework where the LLM itself learns to perform document selection and answer generation in an end-to-end manner. Inspired by human reasoning, our model organizes documents via hierarchical semantic IDs and selects relevant content through a self-reflection mechanism composed of query-specific attention and an additional feed-forward MLP layer. This architecture enables the model to promote helpful documents directly during generation, eliminating the need for separate retrievers or rerankers. Through joint training, the model learns to select the most informative 2-3 documents. We conduct experiments to validate the effectiveness of our design.

## 1 Introduction

Large language models (LLMs) have demonstrated remarkable capabilities across a wide range of natural language processing tasks, including open-ended conversation (Wang et al., 2024a; Liu et al., 2024; Xi et al., 2025), problem solving (Cobbe et al., 2021; Wei et al., 2022; Lewkowycz et al., 2022), code generation (Chen et al., 2021; Austin et al., 2021; Li et al., 2022). Their success is largely attributed to the scale of their architectures and the massive datasets used during pretraining, which allow them to encode vast amounts of linguistic and factual knowledge. However, despite these strengths, LLMs are inherently limited by the static nature of their pretraining corpus. When presented with complex, ambiguous, or unfamiliar queries, especially those requiring recent or specialized knowledge, they often generate inaccurate or fabricated responses, a behavior commonly referred to as hallucination. This issue becomes even more pronounced in domains where information evolves rapidly or factual accuracy is essential.

To address the hallucination problem in large language models (LLMs), Retrieval-Augmented Generation (RAG) (Lewis et al., 2020) has emerged as a practical solution. Rather than relying solely on static knowledge stored in model parameters, RAG enables LLMs to retrieve and condition on external documents at inference time, grounding their responses in up-to-date and domain-specific evidence. Despite its effectiveness, it have several intrinsic drawback.

(1) The performance of retrieval-augmented generation (RAG) relies critically on the quality of the retrieved documents, yet the retriever's objective is often misaligned with the goal of generation. Most retrievers are trained to maximize semantic similarity between a query and candidate documents, typically through embedding-based scoring (Karpukhin et al., 2020). However, such similarity does not guarantee that the document contains factual or contextually useful information for answering the query. One may expect these semantically similar yet uninformative documents to be harmless. Surprisingly, Cuconasu et al. (2024) show that such documents can actively degrade performance, sometimes performing worse than inserting random documents into the prompt. For instance, when answering "Who won the first Nobel Prize in Physics", a misleading document about Einstein may be more harmful than a random one (Jin et al., 2024; Wang et al., 2024b). These find-

ings motivate the exploration of retriever metrics that are more directly aligned with the generation objective.

(2) RAG systems depend on the long-context processing ability of large language models, as multiple retrieved documents are often concatenated into a single input. This not only increases computational overhead but also weakens the model's ability to leverage the retrieved information. As shown by Liu et al. (2023), even models explicitly trained for long-context exhibit performance degradation when relevant information appears in the middle of the input, rather than at the beginning or end. Such positional sensitivity undermines the effectiveness of retrieval and necessitates document reranking. However, as with retrieval, designing reranking objectives that are well aligned with the generation task remains challenging. Moreover, introducing additional components such as rerankers complicates the training pipeline and can lead to training instability.

To address these limitations, we adopt an intuitive strategy: allow the language model itself to decide which documents are most useful for answering a given question. Rather than depending on a retriever with manually crafted heuristics or embedding similarity, we design the model to learn document selection directly from data in a hierarchical manner. This idea is inspired by how humans naturally approach complex questions: they first organize available information into conceptual or topical categories, and then selectively search within those categories for the most pertinent details. By mimicking this behavior, we encourage the model to develop a coarse-to-fine understanding of the corpus, leveraging hierarchical semantic ids, a concept borrowed for generative retrieval (Wang et al., 2022c), to discriminate between document clusters and select those most likely to support accurate generation.

To support this process, we introduce a lightweight self-reflection mechanism that plays a central role in enabling the model to perform effective document selection. This mechanism consists of additional query-specific attention heads and an independent MLP layer, designed to help the model leverage its intrinsic knowledge and internal representations when selecting candidate documents. Built upon this structure, we train the model in an end-to-end manner, allowing it to jointly learn both document selection and answer generation. For each query, the model first identifies a set of candidate documents, each annotated with hierarchical IDs that reflect their semantic or topical structure. It then selectively incorporates the most relevant candidates into the generation process, learning to associate specific hierarchical patterns with successful answer outcomes. Through this training paradigm, the model effectively aligns document selection with downstream generation quality, without relying on external retrievers or manual scoring heuristics.

Unlike traditional RAG pipelines, which decouple retrieval, reranking, and generation into distinct modules, our framework unifies these components within a single model. The language model simultaneously acts as retriever, reranker, and generator, leveraging the same set of internal parameters across all stages (See Figure 1). This design eliminates the need for an explicit reranking step. Through end-to-end fine-tuning, the model learns to promote the most helpful documents to the top of its input sequence, effectively aligning document selection with the downstream generation objective. Meanwhile, documents that are unhelpful or distracting are implicitly filtered out during training, as their lack of contribution to generation quality provides a negative learning signal. This tightly coupled optimization enables the model to perform competitively even when selecting only the top 2-3 documents.

Our contributions can be summarized as follows:

- We propose an end-to-end framework that enables a large language model to jointly perform document selection and answer generation without relying on external retrievers or rerankers. By leveraging hierarchical semantic identifiers inspired by generative retrieval, the model learns a coarse-to-fine understanding of the document corpus and selects evidence that directly supports the generation task.

- To enable large language models to retrieve documents, a capability not acquired during pretraining, we introduce a lightweight self-reflection module composed of query-specific attention heads and an auxiliary MLP layer. This component allows the model to internalize relevance judgments that are synchronized with generation utility, effectively eliminating the misalignment between the retrieval and generation modules.

- Through unified training, the model learns to surface the most helpful documents and discard unhelpful ones, achieving strong performance while conditioning on only the top 2-3 selected inputs.

Experimental results show that our method outperforms traditional RAG methods, demonstrating both efficiency and effectiveness.

## 2 RELATED WORK

**Information Retrieval (IR).** Information retrieval techniques aim to efficiently obtain, process, and interpret information from large-scale data. Traditional approaches, known as sparse retrieval, enable fast document search through inverted indexing, where each term is mapped to a list of documents containing that term. Relevance is then determined using term-matching metrics such as TF–IDF (Ramos et al.), query likelihood (Lafferty & Zhai, 2001), and BM25 (Robertson et al., 2009). With the development of pre-trained language models (Devlin et al., 2019; Liu et al., 2019), several works have leveraged Transformer-based encoders to generate dense vector representations for both queries and documents, with similarity typically measured using inner product or cosine similarity (Karpukhin et al., 2020; Xiong et al., 2020; Wang et al., 2022b;a). In contrast, generative retrieval (De Cao et al., 2020; Tay et al., 2022; Wang et al., 2022c; Zhou et al., 2022) represents a different paradigm: instead of relying on similarity matching in the embedding space, it takes the query as input and directly generates document identifiers (DocIDs) corresponding to relevant documents. Specifically, each document in the corpus is assigned a unique identifier, and the retrieval model employs constrained beam search to ensure that the generated DocIDs correspond to valid documents within the corpus. More recent works have focused on the retriever model training (Zhou et al., 2023; 2024), the construction of semantic identifiers (Sun et al., 2023; Yang et al., 2023; Askari et al., 2024; Valluri et al., 2024), and continual learning on dynamic corpora (Mehta et al., 2022; Kishore et al., 2023; Guo et al., 2024).

**Retrieval Augmented Generation (RAG).** Early efforts to integrate retrieval mechanisms for improving text generation quality can be traced back to Chen et al. (2017); Dinan et al. (2018); Weston et al. (2018). In particular, retrieval systems play a crucial role in open-domain question answering, where a two-stage framework is commonly adopted: a context retriever first selects a small subset of passages, some of which may contain the answer to the question, and a generator then identifies the correct answer from these passages (Chen et al., 2017). Subsequent research has focused on improving retrieval quality by employing dense representing vectors (Karpukhin et al., 2020), or combining the masked language model (Devlin et al., 2019) with the retrieval system (Lee et al., 2019; Guu et al., 2020). This line of work was later formalized under the term Retrieval-Augmented Generation (RAG) by Lewis et al. (2020), which generalizes the framework to all sequence-to-sequence models. After large language models with billions of parameters emerged and demonstrated their superior performance in language generation, further studies explored how RAG could be leveraged to strengthen these models (Izacard & Grave, 2020; Borgeaud et al., 2022; Jiang et al., 2022). Numerous studies have proposed methods that focused on different aspects of retrieval-augmented generation, including training of retrievers or generators (Weijia et al., 2023; Izacard et al., 2023; Lin et al., 2023; Li et al., 2024), instruction fine-tuning (Wang et al., 2023), leveraging in-context abilities (Huang et al., 2023; Trivedi et al., 2022; Wang et al., 2024c), adaptive document selection (Jiang et al., 2023; Asai et al., 2024; Yan et al., 2024; Su et al., 2024; Baek et al., 2024; Jeong et al., 2024; Wang et al., 2024b), passage ranking (Yu et al., 2024), context compressing (Xu et al., 2024a), and parametric knowledge injection (Su et al., 2025). In addition, several studies have investigated how to retrieve relational knowledge relevant to a given query from a pre-constructed graph database (Edge et al., 2024; Hu et al., 2024; Mavromatis & Karypis, 2024; Peng et al., 2024).

## 3 PRELIMINARIES

In this section, we present the framework for reasoning over a document corpus. We consider the standard QA task, where we take in a query $x \in \mathcal{X}$, and an LLM, denoted by $p_\theta(\cdot|x)$, which outputs a conditional probability distribution over the answer space $\mathcal{Y}$ and generates an answer $a$ by sampling from this distribution. Let the QA dataset be denoted by $\mathcal{D}_{\text{QA}} = \{(x_i, a_i)\}_{i=1}^M$, we consider the following training loss of log-likelihood, where each $(x_i, a_i)$ is a query-action pair. The standard training objective is to maximize the log-likelihood of the ground-truth answers under the model,

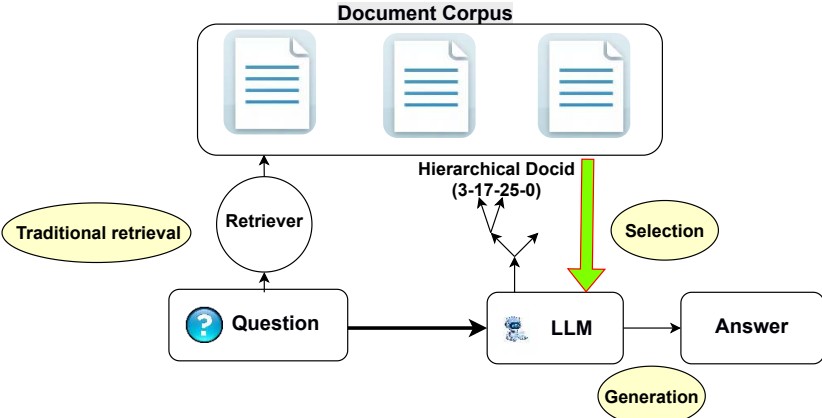

Figure 1: Comparison between the traditional retrieval process and our proposed pipeline. In the traditional setup, the retriever and generator are independent modules: the retriever first ranks documents, and the generator conditions on the top-$k$ results. In contrast, our method enables the LLM to generate hierarchical docids, which directly identify the most relevant document in a semantically structured space.

leading to the loss function

$$L_{\text{QA}}(\phi) = -\sum_{i=1}^{M} \left[ \sum_{j=1}^{m} \log p_\theta\big(a_i^j \mid x, a_i^{[1:j-1]}\big) \right], \tag{3.1}$$

where $a_i^j$ is the $j$-th token of $a_i$. Although fine-tuning the LLM by minimizing this loss function is straightforward, it may fail to yield improvements when the QA task requires knowledge absent from the pretrained model. In such cases, the model parameters, constrained to remain close to their pretrained values, cannot adequately capture the missing information, leading to persistently high loss and consequently little or no improvement in the quality of the generated answers.

To mitigate this problem, we assume access to an external corpus $\mathcal{D} = \{d_1, d_2, \ldots, d_N\}$. We further assume that the knowledge required to answer the question can be found in the document corpus. In the following, we will first review the RALM pipeline, which serves as a baseline framework for incorporating external knowledge into the generation process.

**Retrieval augmented generation (RAG).** In RAG, a retriever model $g_\phi(\cdot|x)$, parametrized by $\phi$, is employed to select a subset of documents $\mathcal{D}^* \subseteq \mathcal{D}$ given the query $x$. For a detailed survey of retrieval approaches, we refer the reader to Section 2. The retrieved documents $\mathcal{D}^*$ is then provided, together with the query $x$, as input to the generator $p_\theta$, which produces an answer by sampling from the conditional distribution

$$\widetilde{a} \sim p_\theta(\cdot|x, \mathcal{D}^*).$$

When $\mathcal{D}^*$ contains the supporting evidence, the distribution is expected to generate answers of higher quality compared to the distribution conditioned solely on $x$.

However, retrieving documents that are genuinely helpful for generation remains challenging. Since most retrieval methods are optimized for semantic similarity rather than generation task, they may return documents that are topically related yet uninformative for answering the query, with their actual contribution to generation becoming apparent only after being processed by the generator. This limitation is exacerbated when fine-tuning the language model, as minimizing the retrieval-augmented QA loss

$$L_{\text{QA}}^{\text{RAG}}(\theta) = -\sum_i \log p_\theta(a_i|x_i, \mathcal{D}^*)$$

depends critically on the quality of the retrieved documents. When performance fails to improve, it becomes unclear whether the bottleneck arises from the quality of the retrieved documents or from

the generator's ability to effectively utilize them during answer generation. While some studies have proposed joint training strategies that simultaneously optimize retriever parameter $\phi$ and generator parameter $\theta$, the two components maintain fully independent hidden representations, with no parameter sharing or representational alignment. Consequently, even when optimized together, the training process fails to capture or exploit the inherent relatedness between retrieval and generation, thereby limiting the potential gains from joint learning.

# 4 METHODOLOGY

## 4.1 HIERARCHICAL SEMANTIC IDS

We apply the idea first proposed in (Wang et al., 2022c) to label the document with hierarchical docids. Specifically, this method assigns each document a structured semantic identifier (docID) that reflects its position within a tree of semantic clusters. Specifically, documents are first embedded into vector representations using a pretrained encoder such as BERT. These embeddings are then clustered using the $k$-means algorithm. If a cluster contains more than a predefined threshold $c$ of documents, $k$-means is applied recursively to produce finer-grained clusters. This process continues until every leaf node contains at most $c$ documents.

This hierarchical id exhibits two notable properties that are important for our design. First, documents with longer common prefixes in their semantic identifiers tend to be semantically similar. This means that the identifier structure encodes coarse-to-fine semantic relationships: documents grouped under the same high-level cluster share the initial segments of their identifiers, while finer distinctions emerge in later segments. As a result, the model can leverage this structure to better locate and discriminate between relevant documents based on shared semantics.

Second, the semantic identifiers can be generated in an autoregressive manner. Specifically, given a document identifier $i_d = [i_1, i_2, \ldots, i_m]$, we can apply an autoregressive retriever model $g_\phi(\cdot|x)$ to generate its docids, to predict the sequence of indices one step at a time, conditioned on the query $x$ and the previously predicted indices. To train the model, we minimize a retrieval loss similar to the supervised fine-tuning loss used in question answering (3.1), by minimizing the following retrieval loss:

$$L_{\text{retriever}}(\phi) = -\mathbb{E}\left[\sum_{j=1}^{m} \log g_\phi\big(i_j \mid x, i_{[1:j-1]}\big)\right]. \tag{4.1}$$

The structural similarity between Equation (3.1) and Equation (4.1) motivates our design to unify retrieval and generation into a single language model, streamlining both stages under a shared architecture and training objective.

## 4.2 UNIFYING RETRIEVAL AND GENERATION WITH A SINGLE LLM

Our approach draws on the parallel between generative retrieval and autoregressive generation in large language models (LLMs). We begin with a standard decoder-only Transformer architecture, where hidden representations are iteratively updated using self-attention and feedforward layers across multiple layers. Each layer processes hidden states $\mathbf{H} = [\mathbf{h}_1, \ldots, \mathbf{h}_T] \in \mathbb{R}^{T \times d}$ through a self-attention mechanism, followed by a gated feedforward network, allowing the model to capture contextual dependencies and non-linear interactions.

In the standard self-attention mechanism, each token in the sequence is projected into query ($Q$), key ($K$), and value ($V$) vectors using learned linear projections, i.e.,

$$\mathbf{Q} = \mathbf{H}\mathbf{W}_Q, \mathbf{K} = \mathbf{H}\mathbf{W}_K, \mathbf{V} = \mathbf{H}\mathbf{W}_V,$$

where $\mathbf{H}_Q, \mathbf{H}_K, \mathbf{H}_V$ are the learned projection parameters. These projections allow the model to compute similarity scores between queries and keys, which are then used to weight the values and produce context-aware representations. This self-attention mechanism, defined as:

$$\text{Attn}(\mathbf{H}) = \text{softmax}\left(\frac{\mathbf{Q}\mathbf{K}^\top}{\sqrt{d_k}}\right)\mathbf{V},$$

serves as the core operation for computing contextualized representations by dynamically weighting token-to-token interactions, thereby enabling the model to capture both local and global dependencies across the sequence.

When extending language models to perform document retrieval, the conventional attention heads, originally optimized for natural language generation, may lack the inductive bias and representational capacity necessary to distinguish useful documents from irrelevant ones. This limitation arises because generation-focused queries are trained to capture linguistic fluency and token dependencies, rather than evidence relevance. Nonetheless, retrieval and generation typically share a common input prefix, the question. The semantic understanding of the question is encoded into key and value representations, forming a latent memory stored in the attention cache and reused across all heads. Although these key-value (KV) pairs are designed to support next-token prediction for question answering, they also contain rich semantic information that can be repurposed for document selection.

To make use of the shared query understanding stored in the attention, we introduce a dedicated retrieval pathway within the transformer architecture by adding a separate set of query projection heads specifically designed for document selection. The retrieval-specific projection, denoted by $\mathbf{W}'_Q$, produce a new set of queries $\mathbf{Q}' = \mathbf{H}\mathbf{W}'_Q$ from the hidden representations $\mathbf{H}$. Unlike the standard query projections used for generation—which prioritize syntactic fluency—these retrieval queries are optimized to evaluate semantic relevance between the query and candidate documents. Importantly, $\mathbf{Q}'$ interacts with the same key–value (KV) pairs, $\mathbf{K} = \mathbf{H}\mathbf{W}_K$ and $\mathbf{V} = \mathbf{H}\mathbf{W}_V$, as in the main attention stream. These KV pairs encode the shared semantic information extracted from the input prefix (the query), serving as a latent memory available to all heads. The retrieval attention is computed as:

$$\text{Attn}'(\mathbf{H}) = \text{softmax}\left(\frac{\mathbf{Q}'\mathbf{K}^\top}{\sqrt{d_k}}\right)\mathbf{V}.$$

Figure 2: Transformer block design for unified retrieval and generation. Left stream for retrieval. Right stream for the original generation.

To further process the retrieval-specific attention output, we introduce an additional MLP layer that mirrors the standard feedforward network used in Transformer blocks. Rather than feeding this into the same MLP used for generation, we apply an additional retrieval-specific MLP, defined as:

$$\mathbf{H}'_{\text{out}} = \text{MLP}'(\text{Attn}'(\mathbf{H})).$$

For comparison, the standard generative path proceeds as:

$$\mathbf{H}_{\text{out}} = \text{MLP}(\text{Attn}(\mathbf{H})).$$

This design decouples the computation paths for generation and retrieval, allowing the retrieval-specific MLP to specialize in evaluating document relevance without interfering with generation fluency. After processing, the retrieval-enhanced output is integrated back into the main hidden state stream via a residual connection:

$$\mathbf{H}_{\text{out}} = \mathbf{H}_{\text{out}} + \mathbf{H}'_{\text{out}}. \tag{4.2}$$

Finally, we introduce a decoder head that projects the final hidden states into a vocabulary space tailored for document identifiers. This projection produces logits over the docid token space, enabling the model to autoregressively generate semantic document identifiers. With this addition, our framework equips the original language model with retrieval capabilities, allowing it to perform document selection through docid generation. Finally, we train the new added parameters by optimizing

$$L_{\text{retriever}}(\phi) = -\mathbb{E}\left[\sum_{j=1}^{m} \log p_{\theta,\phi}^1\big(i_j \mid x, i_{[1:j-1]}\big)\right], \tag{4.3}$$

Here, $\theta$ denotes the parameters of the original language model, while $\phi$ represents the newly introduced parameters for the retrieval component. We distinguish between two output modes depending on whether retrieval is active. In the **first** output mode, denoted as $p^1_{\theta,\phi}$, the model generates document identifiers from the docid vocabulary space. During the pretraining of $\phi$, we freeze $\theta$, ensuring that the retrieval module learns independently without degrading the generation ability of the base language model. When the residual connection in (4.2) is blocked, the system operates solely as a generator, behaving identically to the original language model. However, when (4.2) is enabled, the system transitions into the **second** output mode, denoted as $p^2_{\theta,\phi}$, where document retrieval and answer generation are combined. In this case, the retrieved documents can be incorporated into the generation process, enabling the model to produce answers that are both contextually grounded and faithful to the retrieved evidence.

### 4.3 JOINT PROCESS WITH CROSS ATTENTION

In the previous section, we have described our unified framework for retrieval and generation: the model performs retrieval when the residual path in Equation (4.2) is active, and generation when this path is blocked. However, even under this unified setting, the two components—retrieval and generation—operate in parallel, without direct interaction. This limits the ability of the generation process to dynamically leverage evidence surfaced during retrieval. To more tightly couple the two, we introduce a cross-attention layer that explicitly bridges their hidden states. To be more specific, we consider

$$\mathbf{H}_{\text{out}} = \mathbf{H}_{\text{out}} + \mathbf{H}'_{\text{out}} + \text{CrossAttention}(\mathbf{H}_{\text{out}}, \mathbf{H}'_{\text{out}}), \tag{4.4}$$

where the cross attention function is defined as:

$$\text{CrossAttention}(\mathbf{H}_{\text{out}}, \mathbf{H}'_{\text{out}}) = \text{softmax}\left(\frac{\mathbf{Q}'_{\text{out}}\mathbf{K}_{\text{out}}^{\top}}{\sqrt{d_k}}\right)\mathbf{V}_{\text{out}},$$

with $\mathbf{Q}'_{\text{out}} = \mathbf{H}'_{\text{out}}\mathbf{W}^Q_{\text{out}}$, $\mathbf{K}_{\text{out}} = \mathbf{H}_{\text{out}}\mathbf{W}^K_{\text{out}}$, $\mathbf{V}_{\text{out}} = \mathbf{H}_{\text{out}}\mathbf{W}^V_{\text{out}}$. This design provides a direct communication channel between the retrieval-enhanced representations and the generation stream. Unlike independent processing, the retrieval pathway contributes relevance-aware signals that guide generation, and generation in turn reinforces which aspects of retrieval are most useful. This mutual interaction forms a tight bridge between the two processes, enabling the model to more effectively ground its responses in selected documents without losing fluency.

### 4.4 END-TO-END JOINT TRAINING

With all the structural designs described above, we are now able to conduct end-to-end joint training of the complete retrieval–generation system. In a complete single training step, the model first performs a forward pass through the retriever to select the document identifiers most relevant to the query. By looking up the corresponding document contents, we then concatenate them with the query and perform a second forward pass through the generator to produce the final answer. This two-stage design unifies retrieval and generation in a differentiable pipeline, enabling shared optimization of both components.

Different from prior work, our framework naturally supports two distinct training modes.

**Mode 1:** When the ground-truth document id $i_d = [i_1, i_2, \ldots, i_m]$ is available, we can minimize the loss incurred by the first forward process, which directly supervises the retriever and generator jointly from the retriever perspective:

$$L^1_{\text{joint}}(\theta, \phi) = -\mathbb{E}\left[\sum_{j=1}^{m} \log p^1_{\theta,\phi}\left(a_i^j \mid x, a_i^{[1:j-1]}\right)\right] \tag{4.5}$$

**Mode 2:** On the other hand, when explicit document evidence is not available in the training data, we rely on the retrieval component to propose relevant candidates. In this case, we first sample document paragraphs using $p^1_{\theta,\phi}$ and then evaluate the model directly through the end-to-end QA generation process. The loss is thus computed with respect to the final answer generation, conditioned on both the query $x$ and the retrieved documents $\mathcal{D}^*$:

$$L^2_{\text{joint}}(\theta, \phi) = -\mathbb{E}\left[\sum_{j=1}^{m} \log p^2_{\theta,\phi}\left(a_i^j \mid x, \mathcal{D}^*, a_i^{[1:j-1]}\right)\right]. \tag{4.6}$$

This formulation ensures that the parameters $\phi, \theta$ are trained not only to approximate ground-truth docids, but also to optimize for downstream QA performance. In other words, the retriever is rewarded for selecting documents that lead to better answers under the joint distribution $p_{\theta,\phi}^2$.

## 5 EXPERIMENTS

### 5.1 EXPERIMENT SETUP

In our experiment, we evaluate our model in commonly used open-QA datasets.

**Natural Questions (NQ)**. The NQ dataset (Kwiatkowski et al., 2019), is built from real, anonymized, and aggregated queries issued to the Google search engine, paired with corresponding Wikipedia pages. Each example contains a natural user query along with a human-annotated answer span, which may be either long or short. In our experiments, we leverage the query–document correspondence data for retrieval warm-up training. For the QA task, we adopt the open-domain version (NQ-Open) introduced by Lee et al. (2019), where only questions with short-form answers are retained, and models must retrieve the supporting evidence from the full Wikipedia corpus.

**TriviaQA.** The TriviaQA dataset (Joshi et al., 2017) is a reading comprehension dataset. The questions are authored by Trivia enthusiasts, forming natural question–answer pairs. On average, each question is associated with six supporting evidence documents, which are collected retrospectively from both Wikipedia and the Web. A single query in TriviaQA may correspond to multiple reference documents and multiple valid answers.

For the large language model backbone, we select two competitive open-source models for the generation task: Llama-3.1-8B-Instruct (Dubey et al., 2024) and Qwen3-4B-Instruct-2507 (Yang et al., 2025). We evaluate these models under three different settings:

- No RAG: The model answers questions without retrieval augmentation.

- Vanilla RAG: Use dense passage retriever (DPR) (Karpukhin et al., 2020) for retrieval.

- Our method: Unify retrieval and generation and do joint training.

In our method, we divide the training pipeline into two different stages.

### 5.2 TRAINING RETRIEVAL

We begin by conducting a warm-up training stage for the retriever, where we minimize the objective in (4.5). To represent document identifiers, we reserve a set of special tokens and initialize their embeddings by copying from existing token embeddings. In addition, we introduce a retrieval marker token, <retrieve_token>, which signals the start of docid generation. This warm-up phase ensures that the retriever learns to associate queries with their corresponding documents based on the training data format, thereby guaranteeing that the retrieved documents are meaningful and relevant. We close the cross-attention in this stage.

### 5.3 TRAINING GENERATOR

During generator training, we augment the base model with a LoRA structure (Hu et al., 2022), enabling parameter-efficient adaptation. In this stage, we jointly optimize the newly introduced parameters for the retriever, the cross-attention module, and the LoRA components. To preserve retrieval capability and prevent degradation, we train on a balanced mixture of data from both mode 1 (retrieval-focused) and mode 2 (generation-focused). Consequently, the overall training objective becomes a weighted combination of the two losses, (4.5) and (4.6). This design is feasible because the retriever and generator share parameters, allowing both components to reinforce each other while maintaining consistency across tasks. Moreover, we introduce a generation marker token, <generates_token>, which signals the start of answer generation.

Table 1: Evaluation of our method on open-QA datasets

| Model | Method | NQ | TriviaQA |
|---|---|---|---|
| Llama3.1-8B-Instruct | No RAG | 28.8 | 62.0 |
| | Vanilla RAG | 47.7 | 64.1 |
| | Our method | **51.3** | **71.9** |
| Qwen3-4B-Instruct | No RAG | 29.3 | **57.5** |
| | Vanilla RAG | 40.8 | 47.4 |
| | Our method | **43.2** | 47.9 |

## 5.4 EXPERIMENT RESULTS

We present our experimental results in Table 1. Overall, the results demonstrate that our method can be regarded as a more effective retrieval-injection strategy compared with the baselines. The performance improvement is more pronounced on NQ than on TriviaQA. One possible reason is that TriviaQA often contains multiple valid answers per query, whereas our training setup only uses the first annotated answer as the label to reduce computational cost.

## 6 DISCUSSION

The central idea of this paper is to unify retrieval and generation within a single model, mirroring the natural way humans consult documents: we recall potential sources, select the most relevant ones, and integrate them directly into reasoning. We believe that this joint perspective is also theoretically meaningful from an information-theoretic standpoint. In particular, Xu et al. (2024b) character-ize large language models as performing latent variable inference. Given a prefix $x, a_{[1:i-1]}$, the probability of generating the next token $a_i$ can be described as

$$p\big(a_i|x, a_{[i-1]}\big) = \int_{\mathcal{Z}} p(a_i|x, a_{[1:i-1]}, z) \cdot p(z|x, a_{[1:i-1]}) \mathrm{d}z,$$

where $\mathcal{Z}$ is the space of high dimensional concept variable. Given a set of evidence documents $\mathcal{D}^*$, the probability distribution shifts accordingly, altering both the conditional likelihood of tokens and the posterior over latent concepts:

$$p\big(a_i|x, \mathcal{D}^*, a_{[i-1]}\big) = \int_{\mathcal{Z}} \underbrace{p(a_i|x, \mathcal{D}^*, a_{[1:i-1]}, z)}_{I_1} \cdot \underbrace{p(z|x, \mathcal{D}^*, a_{[1:i-1]})}_{I_2} \mathrm{d}z.$$

Following Xu et al. (2024b), using the Bayesian formula, we can represent $I_2$ as

$$I_2 \propto \underbrace{p(\mathcal{D}^*, a_{[1:i-1]}|x, z)}_{I_3} \cdot p(z|x).$$

Therefore, with some further analysis, Xu et al. (2024b) explained the benefit and detriment of re-trieval augmented generation as distribution completion and distribution contradiction. We extend this view by analyzing how end-to-end training can optimize these distributions with data. In partic-ular, we assume that $p(z|x)$ remains fixed, as it is primarily determined during large-scale pretrain-ing and reflects the intrinsic concept distribution given the input. Consequently, the optimization in downstream tasks focuses on the shifted terms. Specifically, $I_1$ captures the probability of generat-ing the correct answer conditioned on both the evidence and the latent concept, while $I_2$ governs the ability to retrieve appropriate documents by shaping the posterior over concepts. This aligns with our intuition that it is preferable to let the language model select the supporting documents by itself as the retrieval process becomes an internal component of the model's inference.

## REPRODUCIBILITY STATEMENT

In Section 4, we provide a detailed description of the model architecture used in this paper. Additional implementation and experimental details can be found in Appendix A.

## ETHICS STATEMENT

In this paper, we employ large language models (LLMs) to address standard open-domain question answering (QA) tasks. All documents used are drawn from widely adopted benchmark datasets, primarily consisting of Wikipedia articles and other reputable web sources. During the training process, the LLM does not produce any harmful content, including discriminatory, biased, or unfair outputs. As a result, this work does not raise any ethical concerns related to data usage or model behavior.

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

## A    EXPERIMENT DETAILS

For the doc-id generation, we directly follow the construction of Wang et al. (2022c). In the model architecture, we incorporate four additional query heads, denoted as $Q'$, in the attention block of each layer.

For the retrieval training phase, the model is trained on 8 NVIDIA A100 GPUs, each with 80 GB of memory, and the process takes approximately 4 hours. The learning rate is set to $2 \times 10^{-4}$. In contrast, the generation training phase is more computationally intensive, requiring around 8 hours on the same hardware configuration. A smaller learning rate of $1 \times 10^{-5}$ is used to ensure stable fine-tuning of the generator module.

To assess the performance of our system, we evaluate whether any of the gold answers is found within the generated output. Since in our methods, we do not apply complicated prompts, but a single token for the task identification, we utilize a basic input template that explicitly separates the query and retrieved context for fairness. The format is as follows:"Q: {query}\n\nContext:\n{context}A:".

# B   THE USE OF LARGE LANGUAGE MODELS (LLM)

We leverage large language models (LLMs) to structure our ideas and assist in writing logically organized and coherent paragraphs.

