# OpenReview forum: "Retrieval as Reasoning: Learning to Select and Generate with LLMs"
_ICLR.cc/2026/Conference — Submitted to ICLR 2026_

### Official Review · Reviewer_vEmo · 2025-10-27

**Soundness:** 2
**Presentation:** 2
**Contribution:** 2
**Rating:** 2
**Confidence:** 4

**Summary:**

This paper proposes an end-to-end retrieval-augmented generation framework that addresses key limitations of traditional RAG systems—misaligned retrieval and long-prompt inefficiency—by enabling the LLM to learn document selection and generation jointly. Inspired by human reasoning, the model uses hierarchical semantic IDs and a self-reflection mechanism to dynamically identify and prioritize relevant documents during generation, eliminating external retrievers or rerankers. Experiments show the model effectively selects 2–3 informative documents, improving efficiency and performance.

**Strengths:**

* The paper is easy to follow
* Trying to remove the reranker is interesting

**Weaknesses:**

* The novelty is limited, the authors seem only try use generative retriever instead of vanilla retriever, but this has already been well studied
* In the experiments, the authors compare their method with vanilla RAG, but their method uses a trained generator, while vanilla RAG simply use the original model, this seems to be an unfair comparision
* The authors primarily focus on the model training process; however, in my view, the most intriguing aspect lies in how the document tree structure is organized and how the docIDs are generated. But, the paper simply use Bert and K means to conduct clustering, which is trivial.

**Questions:**

* How is the semantic ID organized, why it can represent the position within a tree of semantic clusters
* The authors claim that rerankers are not needed with their unified model, but it seems that the model can only act as retriever and generator, an extra reranker should still help the performance.
* I want to know the performance of the trained generative retriever, is it better than vanilla retrievers like BM25 and contriever

---

### Official Review · Reviewer_2e8q · 2025-10-28

**Soundness:** 2
**Presentation:** 2
**Contribution:** 2
**Rating:** 2
**Confidence:** 4

**Summary:**

This paper proposes a unified framework that addresses the misalignment between retrieval objectives and generation tasks, and mitigates the issues caused by long prompts, which strain model capacity and introduce positional biases. Specifically, this paper enables the LLM to jointly learn document selection and answer generation, and further guides it to select the 2-3 most informative documents to address the limitations.

**Strengths:**

1. This paper uses LLM as retriever to addresses the misalignment between retrieval objectives and generation tasks.

2. The tree of semantic clusters leverages coarse-to-fine semantic relationships to facilitate document selection.

**Weaknesses:**

1. There are many works for addressing the misalignment between retrieval objectives and generation tasks. For example, R2AG [1] addresses the semantic gap that exists between LLMs and retrievers. The authors should take them into related work or experiments.

2. There are many works that aim to solve the lost-in-the-middle problem in RAG, such as PAM QA [2]. The paper’s claim that selecting only the two to three most informative documents alleviates capacity limits and positional bias is insufficiently substantiated. When multiple top-k documents are relevant to the generation task, the model should attend to all of them rather than truncating the context to just 2–3.

3. There is a lack of sufficient baselines.  This paper's baselines only include No RAG and Vanilla RAG. It is beneficial to include more related baselines, such as R2AG[1], Self-RAG[3] and Adaptive-RAG[4].

4. No ablation studies are provided to quantify the individual contributions of hierarchical docIDs, self-reflection mechanism, or cross-attention components.

5. The figures are unclear and thus confusing. Figure 1 fails to distinguish the proposed method from traditional approaches. In Figure 2, although the Transformer block is intended to clarify the LLM’s retrieval and generation logic, the directions of the arrows are ambiguous.

[1] R2AG: Incorporating Retrieval Information into Retrieval Augmented Generation, EMNLP 2024.

[2] Never Lost in the Middle: Mastering Long-Context Question Answering with Position-Agnostic Decompositional Training, ACL 2024.

[3] Self-RAG: Learning to Retrieve, Generate, and Critique through Self-Reflection, ICLR 2024.

[4] Adaptive-RAG: Learning to Adapt Retrieval-Augmented Large Language Models through Question Complexity, NAACL 2024.

**Questions:**

Please refer to the weaknesses.

---

### Official Review · Reviewer_3zTV · 2025-10-30

**Soundness:** 1
**Presentation:** 2
**Contribution:** 1
**Rating:** 2
**Confidence:** 4

**Summary:**

This paper aims to address key limitations in Retrieval-Augmented Generation (RAG), namely the misalignment between retrieval and generation objectives, and the difficulty models face when processing long input contexts. The authors propose a unified framework where a single Large Language Model (LLM) performs both document selection and answer generation.

The approach utilizes Generative Retrieval (GR), specifically adopting the hierarchical semantic document identifiers (DocIDs) proposed by Wang et al. (2022c). The core technical contribution involves modifying the standard Transformer architecture to include a "self-reflection mechanism." This mechanism introduces a parallel pathway dedicated to retrieval, consisting of specialized query projections (Q') and an independent MLP layer. This retrieval pathway shares the Key (K) and Value (V) representations with the generation pathway. The outputs of the two pathways are combined via a residual connection and an optional cross-attention mechanism.

The model is trained using two objectives: Mode 1 (Eq 4.5), a supervised loss on generating ground-truth DocIDs, and Mode 2 (Eq 4.6), a standard QA loss conditioned on documents retrieved by the model. The authors claim this enables end-to-end joint training, aligning retrieval with the downstream task. Experiments on NQ and TriviaQA using Llama-3.1-8B show improvements over No-RAG and Vanilla RAG (using DPR) baselines, and mixed results on Qwen3-4B.

**Strengths:**

- Well-Motivated Problem: The paper identifies the misalignment between independent retrieval and generation modules as a significant challenge in RAG. The goal of unifying these components into a single, jointly optimized model is a compelling research direction.
- Interesting Architectural Design: The proposed architectural modification (Section 4.2, Figure 2) is interesting. Introducing specialized Q' projections and MLPs for retrieval while reusing the K/V representations from the generation stream is a plausible method for enabling multi-task capabilities within a single decoder architecture.

**Weaknesses:**

- Joint Training Mode 2 (Eq 4.6) Seems Flawed: Eq 4.6 does not learn the retrieval component, but assumes that it could leverage one trained from Mode 1. This assumption might not hold. Even if there is some supervision, there could be domain difference between labeled and unlabeled data. To properly address this problem, some sort of RL / Gumbel relaxation may be needed.
- Missing Generative Retrieval Comparison: As the proposed method is an integration of GR into the generator, a crucial baseline is missing: a standard, standalone GR model (_e.g._, fine-tuning the base LLM to generate DocIDs without the architectural modifications) used as a retriever for the same generator LLM. Without this baseline, it is impossible to determine if the gains come from using GR or from the proposed architectural unification.
- Misleading Framing ("Reasoning"): The title "Retrieval as Reasoning" is not supported by the methodology. The paper describes the approach as inspired by human hierarchical organization of information (L65-74). However, the technical implementation is a mechanism for hierarchical selection via generative retrieval. It does not involve explicit multi-step inference, logical deduction, or iterative refinement typically associated with "reasoning" in the context of LLMs.
- Mixed Results Suggest Expanded Evaluation is Needed: No statistical significance tests are performed. In Table 1, the results for Qwen3-4B on TriviaQA show that Vanilla RAG (47.4) performs significantly worse than No RAG (57.5). The proposed method (47.9) also underperforms the No RAG baseline in this configuration. A more extensive evaluation will help clarify their relative performance.

**Questions:**

- Can you provide ablation studies to isolate the impact of the cross-attention mechanism (Section 4.3) and the dual-stream architecture (Section 4.2)?
- How does the model compare to (a) a RAG pipeline using a state-of-the-art dense retriever and reranker, and (b) a standard Generative Retrieval approach using the same base LLM without architectural modifications?

---

### Official Review · Reviewer_ERwf · 2025-10-31

**Soundness:** 1
**Presentation:** 2
**Contribution:** 1
**Rating:** 2
**Confidence:** 5

**Summary:**

The paper proposes an end-to-end alternative to standard RAG that lets the LLM itself select evidence and generate answers, removing separate retriever and reranker components. It identifies two primary limitations in standard RAG: 1) the misalignment between retrieval objectives and the actual needs of the downstream generation task, and 2) the performance issues arising from processing long contexts (e.g., computational cost and positional bias). To address this, the authors propose a unified, end-to-end framework where the LLM itself learns to perform both document selection and answer generation, thereby replacing external retrievers and rerankers. The core of their method involves organizing documents with hierarchical semantic IDs and integrating a lightweight "self-reflection mechanism" into the LLM. Experiments (briefly described) indicate improved efficiency and accuracy over traditional RAG pipelines under this setting.

**Strengths:**

1. The challenge and problem addressed by this paper is interesting and important.
2. The paper is generally easy to understand.

**Weaknesses:**

This paper seems to be a draft that only basically proves the feasibility of the idea. Several problems should be fixed before being qualified as a top-tier conference paper:

1. Severe lack of experiments: The authors only use Table 1, which consists of two models, two simple qa datasets, and no baselines from other research papers. While the result is better than vanilla RAG, it is far from SOTA in the same setting, according to my knowledge. Besides, since the paper claims the motivations on long documents and reasoning, it should be tested on more complex benchmarks rather than only NQ and TriviaQA.
2. Lack of novelty and references: This idea for unifying retrieval and generation is interesting and worth exploring. However, there are already preceding papers working along this line. For example, GritLM, OneRec/OneRec-Think, UniGen etc. The paper should refer to these works and carefully discriminate between its own contributions.
3. Lack of demonstrations: The figures are too coarse and do not show the importance of key contributions. No case studies can help the readers understand how the model unifies retrieval and generation.

**Questions:**

n/a

---

### Meta-Review · Area_Chair_DheW · 2025-12-24

**Summary:**

The none of reviewers' concerns are addressed.

**Reviewer Concerns:**

The major concerns include (1) the lack of contrasts to directly related papers (e.g., self-RAG) and (2) the lack of empirical justification of the proposed methods on diverse datasets and diverse models, which needs to be improved to meet the bar of top-tier conferences. Non of these concerns are addressed during the rebuttal.

**Reviewer Scores:**

All reviewers agreed to reject this paper initially. As the authors’ didn’t address reviewers’ concerns during the rebuttal period, I clearly expect that all reviewers will maintain their suggestions.

---

### Decision · Program_Chairs · 2026-01-26

Reject